# The Effect of a Resistance Training, Detraining and Retraining Cycle on Postural Stability and Estimated Fall Risk in Institutionalized Older Persons: A 40-Week Intervention

**DOI:** 10.3390/healthcare10050776

**Published:** 2022-04-22

**Authors:** Rafael Nogueira Rodrigues, Eduardo Carballeira, Fernanda Silva, Adriana Caldo-Silva, Cidalina Abreu, Guilherme Eustaquio Furtado, Ana Maria Teixeira

**Affiliations:** 1Faculty of Sport Sciences and Physical Education, Research Center for Sport and Physical Activity (CIDAF, UID/PDT/04213/2020), University of Coimbra, 3040-248 Coimbra, Portugal; geral.fernandasilva@gmail.com (F.S.); dricaldo@gmail.com (A.C.-S.); ateixeira@fcdef.uc.pt (A.M.T.); 2Gerontology & Geriatrics Research Group, Department of Physical Education and Sport, University of A Coruna, 15179 A Coruña, Spain; 3Research Unit of Science of Health (UICISA:E), Nursing School of Coimbra, 3004-011 Coimbra, Portugal; cila@esenfc.pt; 4Research Unit for Inland Development (UDI), Polytechnic Institute of Guarda, 6300-559 Guarda, Portugal

**Keywords:** older adults, postural stability, strength exercise, fall risk, technology-based assessment

## Abstract

Physical inactivity and low levels of muscle strength can lead to the early development of sarcopenia and dynapenia, which may increase the number and risk of falls in the elderly population. Meanwhile, exercise programs can stop or even revert the loss of muscle mass, strength, power, and functional capacity and consequently decrease the risk of falls in older adults. However, there is a lack of studies investigating the effect of strengthening programs in octogenarians. The present study investigates the effects of 40 weeks of a training-detraining-retraining cycle of muscle strength exercise program on postural stability and estimated fall risk in octogenarians. Twenty-seven institutionalized participants were allocated into two groups: the muscular strength exercise group (MSEG, *n* = 14) and control group (CG, *n* = 13). After the first training period, the MSEG improved postural stability and decreased the estimated fall risk by 7.9% compared to baseline. In comparison, CG worsened their stability and increased their risk of falling by more than 17%. No significant changes were found between groups in the detraining and the retraining period. This study demonstrated that strength exercise effectively improved postural control and reduced fall risk scores. In addition, the interventions were able to reduce the forward speed of postural control deterioration in octogenarians, with great increments in the first months of exercise.

## 1. Introduction

Life expectancy is growing worldwide, and consequently, many health problems related to the aging process have been drawing greater attention, with the physical, psychological, and physiological degenerative processes standing out. The increase in sedentarism linked to aging is a global health problem [1]. The lack of physical exercise can trigger the early development of some physical and neurodegenerative diseases in older adults leading to the development of sarcopenia and dynapenia [2,3]. The latter phenomena are part of what has been called the frailty syndrome [4], which has been linked to an increase in the number and severity of falls in older adults [5]. 

The World Health Organization, aware of the importance of the fight against a sedentary lifestyle in the aged population, has published some minimum dose recommendations for health maintenance [6]. These recommendations are to perform at least 150–300 min of moderate physical activity, or 75–150 min of vigorous physical activity per week, including multicomponent exercise focusing on balance and strength, to improve the functional capacity and prevent falls [6].

The fall is conceptualized as an unintentional body displacement to a lower level than the initial position, determined by multifactorial circumstances that compromise stability [7]. Aging plays a central role as it affects the afferent sensory system (i.e., proprioception, vestibular and visual system), the central neurologic control system (i.e., cognition, attention, fear of falling), and finally, the efferent neuromotor system (i.e., physical function, muscle strength, balance, and stability) [8,9]. Changes in central neural system connectivity have been observed in areas related to the integration of information and in areas associated with motor and sensory information processing, providing evidence of the complex multidimensionality of the neural underpinnings of falls [10]. In addition, diseases, drugs, and environment modulate the age-associated changes in the fall risk pathway.

In this scenario, there are intrinsic causes (deterioration of physiological and neuromuscular changes of aging, muscle dysfunctions, pathologies, medications) and extrinsic causes (environmental hazards, architectural and furniture inadequacies, stairs, high heel shoes) of fall occurrence [11]. The interaction between intrinsic and extrinsic factors compromises the perceptive and neuromuscular systems related to postural stability and balance control, affecting the functionality and quality of life of the older adults [12], being an essential aspect of morbidity and mortality and the leading cause of fatal and non-fatal injuries among older adults [13]. 

According to a Behavioral Surveillance and Risk System survey, about 26% of older adults reported falling at least once in the last 12 months, resulting in 24.96 million falls in 2020 [14]; likewise, in 2019 [15], more than 3 million fall injuries and more than 34,000 deaths related to falls were recorded, generating an approximate cost of 50 billion dollars in medical care in the USA [15,16]. Sadly, these numbers kept growing and are projected to increase 30% by 2030, resulting in around seven deaths/hour in the USA [16]. Similarly, in Europe, the Western region saw 8.4 million older adults in medical centers due to fall-related injuries in 2017 [17]. In this scenario, institutionalized older people are at a higher risk of falling since the prevalence of frailty in nursing homes is 50%, and approximately 40% are pre-frail individuals [18]. Furthermore, the percentage of fallers increases from 26.7% in older people between 65 and 74 y/o to 29.8% in older people between 75 and 84 y/o [19]. For people over 84 years old, their incidence of falls increases up to 36.5% [19]. Furthermore, this effect in octogenarians is still not well understood, given this population’s difficulty, specificity, and high vulnerability [20].

Meanwhile, some studies have shown that physical exercise can attenuate the speed of evolution of some neurodegenerative processes, such as sarcopenia, dynapenia, and frailty, contributing to balance control and postural stability [21,22,23]. Regular exercise has been proven to reverse the frailty status and decrease the fall risk among older adults, even in those who live in nursing homes or social care institutions [9,18]. Recent studies have shown that institutionalized older adults had lower scores in physical fitness and higher scores for depressive symptoms and comorbidities, with a significant correlation between frailty, fear, and risk of falling and physical fitness [21,24,25,26,27]. It has been demonstrated that professional-oriented multicomponent training for eight weeks has positive outcomes; specifically, it has been indicated that shorter and high-intensity dynamic exercise can be an effective way of improving performance, gait, and balance capacities in older adults at risk of fall [28].

A meta-analysis has shown that exercise-only interventions had a practical effect on fall risk in institutionalized and non-institutionalized older adults, significantly reducing the number of falls [29]. However, this same meta-analysis showed a high percentage of drop-out ratio in the population studied, making it difficult to draw conclusions from this study. This, taken together with the lack of works focused on people over 80 years old, indicates the need to study the chronic adaptations in very old populations after exercise interventions. The benefits of exercise are transient and last as long as it is being performed; thus, the necessity of adherence and progression is crucial [3,6,18,21]. However, adherence is an unresolved matter when training older persons, especially those institutionalized, who interrupt their training programs, for example, when spending holidays with families and other diverse circumstances. Therefore, the study of evolution/involution of neuromuscular adaptations after training-detraining may help prescribe exercise with more prolonged residual effects to overcome detraining periods [30,31]. In this way, some authors have analyzed the effects of the detraining process [32]. Detraining can be defined as training reduction or cessation, which implies temporary discontinuation or complete abandonment of systematic programmed physical exercise, which may cause a partial or complete loss of training-induced adaptations (anatomical, physiological, psychological, and functional performance) produced during a previous training period [33]. Some authors studying this process have indicated that after 12 weeks of detraining, the benefits of exercise started to decline, and even after another 12 weeks of detraining, some of the muscular endurance and strength parameters reduced by ~15% [30]. In another detraining study, strength and gait speed were reduced after 16 weeks of no training but did not return to their baseline values [34]. These data point to a lasting protective effect from exercise, even when these capabilities decline due to the aging process [35]. In fact, as people age, muscle power deteriorates faster than muscle strength [36].

Muscle strength is the amount of tension that a muscle or muscle group can generate in a specific movement; meanwhile, muscle power is the tension generated at a specific velocity [37]. In this context, neuromuscular adaptations and deteriorations, mainly at the level of muscle-tendon units (i.e., reduced tendon stiffness), muscular structure (i.e., reduced number of muscular units, and atrophy of fast-twitch fibers), and neural changes affect strength capabilities [cite] and power output [38]. However, since muscle power is more strongly associated with daily life activities, more attention must be paid to exercise strategies that contribute to power development in older adults.

In this scenario, the specific use of elastic bands in exercise training programs has been shown to improve muscular capacities such as strength, balance, and functional capabilities in older adults, even in the institutionalized ones [22], including those characterized as frail or pre-frail [23]. Considering the reasons given, we aimed to evaluate the effects of forty weeks of a training-detraining-retraining cycle of muscle strength exercise (MSE) program with elastic bands on institutionalized octogenarians and its influence on postural control and estimated fall risk status.

## 2. Materials and Methods

We have employed a non-probability convenience sampling of octogenarian dwelling older adults living in nursing homes. The institution’s directors and the older adults’ legal representatives revised and signed the consent form before the first testing session. The estimated sample size was calculated using the G*Power software (version 3.1.9.7) [39]. Based on our calculations, for an effect size of 0.30, a sample size of 26 achieves 95% statistical power to detect differences among the means using an ANOVA test with an α-level of 0.05. The sample consisted of 27 participants (7 males, 20 females) aged over 80 (86.37 ± 3.59) years old, institutionalized in nursing homes or social care centers of Coimbra (Portugal). This study is designed as a prospective, naturalistic, controlled clinical trial (treatment vs. care) composed of 3 phases, i.e., training, detraining, and retraining. The participants were stratified randomized into two groups: the muscular strength exercise group (MSEG, *n* = 14, 4 male and 10 female), who performed an elastic band strength exercise program, and the control group (CG, *n* = 13, 3 male and 10 female) who continued their usual routine, which does not include any kind of programmed and supervised physical exercise.

The eligible criteria for the participants in this study were that, at the time of first screening, participants had to be: (i) 80 years old or more; (ii) clinically stable with their drug therapy updated; (iii) not participating in another structured program of physical exercise in the last six months; (iv) not presenting any type of health condition or use medication that might prevent the functional self-sufficiency test performance or attention impairment (such as severe cardiopathy, uncontrolled hypertension, uncontrolled asthmatic bronchitis or severe musculoskeletal conditions); (v) not presenting mental disorders or hearing/visual impairment that could prevent the evaluations and activities proposed, according to the institutional medical staff.

Additionally, we should address that in this study, care was taken to exclude as much as possible the factors that differentiated the sample, seeking a homogeneous sample without statistical differences in age, sex, the ability to perform daily life activities independently, and without a history of diseases that could directly affect the balance so that the results are as accurate as possible.

The intervention consisted of a first period of sixteen weeks of resistance training with elastic bands, followed by eight weeks of detraining and a second training period of sixteen weeks. Participants performed a total of 64 sessions, 2 sessions/week of 45 min each on non-consecutive days distributed between the two training periods.

To avoid any bias, all the participants completed the evaluation protocol at the same time period, between 10 am to 11.45 a.m. That protocol was repeated on four occasions: pre-intervention (PRE), postintervention after 16 weeks of training (POST16), after eight weeks of detraining (POST24), and postintervention after 16 weeks of training in the second period (POST40) (Figure 1). The evaluation protocol assessed anthropometric values and estimated fall risk through an index based on the posturography platform (Physiosensing^®^ v.19002, Sensing Future, Coimbra, Portugal) test with four specific conditions (please see details below).

This study was approved by the Faculty of Sport Science and Physical Education, University of Coimbra Ethics Committee (reference number: CEFCDEF/0028/2018) respecting the Portuguese Resolution (Art. 4th; Law no. 12/2005, 1st series) on ethics in human research [40] and the Helsinki Declaration. Clinical trial register number NCT04376463.

### 2.1. Postural Stability and Fall Risk Assessment

The fall risk assessment allows the identification of potential fall candidates. The protocol employed in the present study, the Physiosensing^®^ Fall Risk test, has been validated and described elsewhere [41,42]. Postural stability assessment was performed employing a specialized force platform (Physiosensing^®^ v.19002, Sensing Future, Coimbra, Portugal) that measured the participants’ sway center of pressure. Each participant stood barefoot on the force platform and tried to be as stable as possible in a static upright position, directing their gaze to a point located at 2 m, for 45 s under four pre-established conditions [41,42]: (1) comfortable stance with eyes open (CSEO); (2) comfortable stance with eyes closed (CSEC); (3) narrow stance with eyes open (NSEO); and (4) narrow stance with eyes closed (NSEC).

Data were stored and analyzed with commercial software (Physiosensing^®^ v.19.0.1.0) that calculated the speed index for each condition to estimate fall risk. The speed index is calculated as the displacement velocity of the center of pressure (i.e., distance traveled in the sagittal plane divided by the test time (mm/s), normalized by the participant’s height, and transformed with the natural logarithm function [1]. The fall risk estimation is based on the assumption that an increment in the participant’s sway velocity denotes a postural control deficit. The software also calculates the composite speed index score as the mean of the scores obtained in the four conditions. The higher scores in the composite index and within each of the four conditions indicate higher fall risk.

### 2.2. Muscular Strength Exercise Protocol

An exercise expert supervised the MSE program. Exercise prescription was based on the recommendations of the American College of Sports Medicine guidelines and previously published exercise prescription guidelines for older adults [43,44]. Furthermore, participants from the MSE group could choose their preferred music to increase the adherence rate [45].

The sessions comprised five minutes of a general warm-up with mobility exercises, and the main part involved resistance exercises with an elastic band for 35 min (see the detailed program in Table 1). At the end of the session, participants completed a cooldown with stretching exercises for 5 min. The program consisted of organized and planned exercises performed with a chair to ensure the safety of the participants. The intensity of the resistance training program was controlled by a rate perceived exertion (RPE) scale (Borg 0–10 scale [46]). During the exercise sessions, participants wore a heart rate monitor (Polar, model M200, Polar Electro Oy, Kempele, Finland), and heart rate (HR) was estimated using Karvonen’s formula where HRmax was calculated using a specific formula for older populations [47]. Heart rate was controlled jointly with the observation of facial flushing or hyperventilation to identify possible adverse events during exercise training. The MSE program consisted of 9 elastic bands exercises per session of progressively increased intensity (TheraBand, Akron, OH, USA). The exercises’ execution targeted truncal musculature, so the proposed exercises, when possible, were executed safely and correctly in stand positions, adding some balance and stability needs, leading to a higher stimulation of the proprioceptive system.

The progression intensity was based on the OMNI table [48], which indicates the intensity progression throughout a colored band progression (soft-to-hard). The exercise protocol consisted initially of 2 sets of 10 to 20 repetitions with a light intensity band during the adaptation period and progressed gradually every 2 to 3 weeks. Finally, for the last four weeks, some free weights (dumbbells and ankle weights) were added to exercises, which allowed for a more intense and diversified spectrum of exercises. The participants performed all exercises in sequential order within each set and employed a cadence of 2 s in the concentric phase and 3 s in the eccentric phase of the movement. A minimum of two days between sessions was provided to ensure sufficient recovery.

The minimum adherence to be considered for analysis was set up to 80% of training sessions. When participants missed two consecutive sessions, a researcher contacted them and offered help to return to the group class; in case of a negative response, they were excluded from the analysis.

### 2.3. Anthropometric Assessment

Participants’ body mass and stature were measured in a portable scale (Seca, model 770, Hamburg, Germany) with 0.1 kg of precision and a portable stadiometer (Seca Body meter^®^, model 208, Hamburg, Germany) with 0.1 cm of precision, respectively. Body mass index (BMI) was calculated as the body mass in kilograms divided by the square of height in meters. Standardized procedures were followed as previously recommended [49].

### 2.4. Statistical Analyses

All descriptive data are presented as estimated marginal means and the 95% confidence interval (CI). Normality was assessed through standard distribution measures, visual inspection of Q–Q plots and box plots, and the Shapiro–Wilk test. Changes within and between groups were analyzed by employing mixed models for repeated measures designs with the module GAMLj [50], which uses the R formulation of random effects as implemented by the function lme4, an R package, in Jamovi software (The jamovi project, v1.6, 2021). GAMLj estimates variance components with restricted (residual) maximum likelihood (REML), producing unbiased estimates of variance and covariance parameters, unlike earlier maximum likelihood estimation. The inter-subject factor group (MSE and CG), the intra-subject factor time (i.e., PRE, POST16, POST24, and POST40) and condition (i.e., CSEO, CSEC, NSEO, NSEC; when appliable), and the interaction (group × time) were set as fixed effects. Sex and age were not introduced as a fixed factor and covariate, respectively, because these variables did not improve the model (i.e., parsimonious method), as evaluated by the Akaike information criterion (AIC). The participants’ intercepts were set as a random effect. Time slope was not included as a random coefficient since this factor’s variance was small in composite scores and speed index by condition.

Within-subject and between-subject changes were evaluated by ANOVA F omnibus test employing the Satterthwaite approximation of degrees of freedom and estimating the coefficients with their 95% confidence intervals for the fixed effects in the mixed model. When a significant interaction was detected, paired and independent comparisons were made with a t-test with the Bonferroni–Holm correction for within-subject and between-group changes, respectively. Furthermore, the variance of the random coefficients was obtained. Simple effects analysis was applied with ANOVA (type III sums of squares) and the Kenward-Roger method for degrees of freedom calculation. The level of significance was established at *p* < 0.05.

## 3. Results

Simple effects analysis revealed that MSEG and CG groups participants did not present significant differences in the anthropometric variables, age, sex distribution, and any postural control conditions and composite index (Table 2) at baseline.

Body mass index did not change from the baseline to POST16, POST24, and POST40 weeks of intervention (*p* = 0.477). The composite index evolution and its percentual delta changes are presented in Table 3. The percentual delta changes (∆%) represent the comparison with the precedent moment of measurement.

In relation to the composite index, we found a main effect of group (F_1,25_ = 7.80, *p* = 0.010), moment (F_3,399_ = 15.15, *p* < 0.001) and interaction of group by moment (F_3,399_= 31.75, *p* < 0.001). Estimated difference between MSEG vs. CG was −2.35 a.u. (95%CI [−4.0 to −0.7]), *p* = 0.010), and the difference between groups (i.e., MSEG vs. CG) in changes of POST16 vs. PRE (β = −2.6, 95%CI = −3.3 to −1.9, *p* < 0.001), POST24 vs. PRE (β = −2.4, 95%CI = −3.0 to −1.7, *p* < 0.001) and POST 40 vs. PRE (β = −2.9, 95%CI = −3.6 to −2.3, *p* < 0.001) indicate lower composite index in MSEG than CG during the entire intervention. Mainly, CG progressively increased their composite index throughout the intervention period (Table 3 and Figure 2).

Mean and individual responses to every moment of postural control conditions (CSEO, CSEC, NSEO, NSEC) are illustrated in Figure 3. We observed a significant difference in postural control between groups (F_1,25_ = 6.44, *p* = 0.018), moments (F_3,375_ = 6.52, *p* < 0.001), conditions (F_3,375_ = 9.16, *p* < 0.001) and moment x group (F_3,375_ = 11.34, *p* < 0.001). There was no triple interaction group x moment x condition (F_9,375_ = 0.53, *p* = 0.853). However, simple effects analysis of group moderated by moment and condition revealed that eyes closed conditions (CSEC and NSEC) showed differences between MSEG and GC (−2.4 a.u., 95%CI = −4.6 to −0.1; and −2.8 a.u., 95%CI = −5.0 to −0.5; respectively), conversely to eyes open condition (CSEO and NSEO). Moreover, differences between groups were evident (*p* < 0.05) in POST24 and POST40 in all conditions. The random intercept (i.e., participants intercept) presented higher variance (σ^2^ = 5.30) than residual variance (σ^2^ = 3.08), which justifies the employment of setting participants as clustered random component. Individual and mean responses can be seen in Figure 3.

## 4. Discussion

The main findings of this study can highlight that (1) 16 weeks of training were sufficient to improve body control in octogenarians, (2) 8 weeks of detraining were sufficient to observe reductions in the improvements see after the training period, but not strong enough do not return to baseline values, (3) retraining process was able to start reverting the reduction seen after detraining, promoting improvements, and (4) the lack of exercise in the CG led to a trend in a decrease/worsen in all body control parameters.

In this way, the main findings of this study can contribute to the growing body of evidence signaling the protective benefits of strength exercise in the very old populations [30,51]. Our results showed that very old adults performing 45 min of muscular strength exercises, two times a week, for 16 weeks could improve stability and body control, represented by a reduction of 7.9% in the composite index. Moreover, even after the performance decline observed after the detraining process, an increment of 6.4% in the composite index was still present, the exercise program showing a protective effect and avoiding returning to baseline values. Additionally, the MSEG increased their stability, reducing 3.1% their composite index, during the retraining process; meanwhile, CG showed a very marked increase of instability after the first 16 weeks (17.1%) followed by a more gradual increase during the subsequent measuring moments (2.4% at POST24, and 2.3% at POST40). These results agree with those reporting an association between lack of physical activity or regular exercise training with a decrement of postural control and a concomitant worsening in the performance of gait and instrumental activities [52,53].

Therefore, exercise programs have shown positive results, especially those focused on increasing strength combined with functional exercise [21,51,54,55]. Accordingly, in a similar design study [28], it has been reported a 14% improvement in balance after a training period, a reduction of stability of 7% after the detraining period, and a new improvement of 18% after retraining. Taken together, these results confirm a positive effect of strength training over the postural control of older adults [56,57]. Moreover, older adults who did not exercise (CG) increased their sway velocity and the composite index, which is interpreted as an increment of their risk of falls.

We also observed that the CG presented a substantial increment of their sway velocity composite index (17.1%) at POST16, when on the other hand, MSEG presented a 7.9% decrease of sway velocity composite index, indicating an improvement in the estimated risk of falls after that period. A similar pattern was observed in the postural control (in the four conditions (Figure 3), with the CG always showing a worsening trend when compared to the MSEG. It has to be emphasized that MSEG trained resistance exercise with elastic bands for two sessions per week, with similar training configurations producing similar results, showing that groups who performed physical exercise demonstrate a trend to decrease de fall risk by improving their balance [24,57].

In this scenario, other studies comparing older adults who practiced physical activities with those who did not practice [54,55] observed that many older adults were prone to falls in both groups. However, those who practiced physical activity regularly showed a higher level of mobility and less propensity to fall when compared to the inactive group.

After the first training period (POST16), we observed the biggest difference between groups in CSEC (*p* < 0.001) and NSEC (*p* < 0.001), these evaluates postural control when visual capacity was taken off, indicating the positive effects of exercise on proprioception, sensorial information, and the vestibular system. Therefore, our results are in line with the study [58] where they found that physical exercise was able to improve significantly the proprioception related to sensorial status in older persons, which is also closely related to the risk of fall [59] and to another study [60] where the practice of exercise apparently was helpful to attenuate the deleterious effect of eye closure on postural control.

After the detraining phase, we expected that the physical capacities of the participants would worsen because of the withdrawal of the exercise program, and consequently, they would diminish their postural control and stability, increasing the risk of fall [19]. However, our study was also able to identify some possible protective effects of exercise because the postural control of the stability test barely varied from POST16 to POST24 in MSEG (0.53%, 0.43%, 0.01%, and 0.09% for CSEO, CSEC, NSEO, and NSEC, respectively). In comparison, percentual increments in instability were observed in CG (10.3%, −1.02%, 9.89, and 0.05% for CSEO, CSEC, NSEO, and NSEC, respectively). These results show a possible protective effect of the exercise program delaying the expected age deterioration of neuromuscular system controlling posture when aging.

In this way, a study [31] observed significant reductions in strength and power after six weeks of detraining, but balance and neuromuscular function did not return to the baseline levels. Therefore, health professionals and researchers have pointed out the relevance of putting their efforts into establishing strategies to delay the aging process or at least maintain the quality of life of people cushioning the consequences of chronic degenerative diseases [22,57,61]. These authors suggested that strength training programs are essential for maintaining muscle strength, balance, functional performance, and independence in older adults.

The results of the present study, when indicating a significant difference in postural stability and fall risk, agree with several studies [24,31,54,62], where physical activity contributes to a lower incidence of falls in older persons. Among the strategies to reduce the action of risk factors for falls, the practice of exercise, like in our study, has been proven as an effective intervention proposal [34,63]. Moreover, most of these studies defend the importance of physical activity and exercise and being active as a method of prevention [6,64,65].

Even if arguing that only the practice of physical activities can reduce the risk of falls by improving body stability and postural control is inconsistent [66,67] because even in the second exercise intervention (retraining), the difference between both groups was not significant, and also, there were cases of older adults in MSEG in whom the score was worse even after the exercise program, this only helps to highlight the multifactorial nature of the process involved. Medications, psychological condition, or even nutritional status, all can affect the risk of fall.

## 5. Conclusions

Our study revealed that sixteen weeks of two 45 min sessions/week of a resistance training program with elastic bands effectively improved balance control and exerted a protective effect reducing fall risk in very old adults (i.e., > 80 y/o).

During the retraining period, both groups did not change significantly in any variable. However, the MSEG obtained better stability outcomes during this period compared to PRE values. Meanwhile, the CG kept a worsening trend during the eight weeks that this period lasted. The delayed beneficial effects produced on the stability of the MSEG group during the first training period meant that at the end of the detraining period, the participants of this group were at a lower risk of falls than the CG. In addition, we should highlight that we observed greater improvements in CSEC and NSEC conditions, where the visual capacity was taken off during the evaluation, showing a possible positive effect of exercise concerning proprioception and the vestibular system, which should be further studied.

Our results indicate that a band-based resistance training program can positively affect postural stability and the risk of falls, providing an increase in balance and postural stability, which can positively ameliorate the functional ability and mobility of older adults. Additionally, the stability trends observed during the detraining period highlight the need to develop better and more specific physical activity programs for very older people (i.e., >80 years) to ensure adherence to training programs and avoid the detrimental effect of being inactive.

Our study has some limitations that should be reported. We did not measure the participants’ daily physical activity levels, which could influence the results. However, since they were older adults living in nursing homes or social care centers, certain stability can be expected in carrying out their usual physical activities since they were prescribed by the different institutions of origin. In this study, we did not control the incidence of falls prospectively in a follow-up period that would allow us to study the effects of the intervention on the incidence of falls. In future studies, it should be considered to include a follow-up period after intervention with different doses of strength training and functional activities to determine those training configurations that most favor the reduction of falls. In addition, other variables that may affect falls (changes in the medication regimen, aspects related to the context of the participants, psychometric evaluations…) must be recorded to analyze holistically the determining factors that reduce the number of falls in octogenarian adults.

## Figures and Tables

**Figure 1 healthcare-10-00776-f001:**
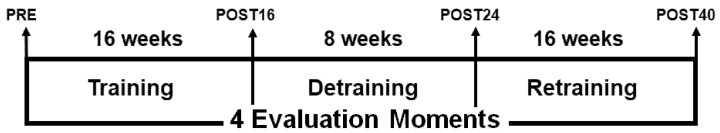
Graphical representation of the study design.

**Figure 2 healthcare-10-00776-f002:**
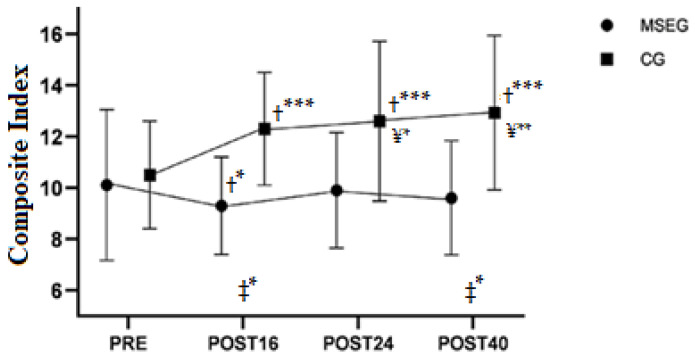
The composite index in the four moments of measurement. PRE: pre-intervention test; POST16: after sixteen weeks of intervention; POST24: after eight weeks of detraining; POST40: after the second training period (retraining). ‡ = difference between groups at that specific time; † = difference versus PRE; ¥ = Difference versus MSEG-POST16. * = *p* < 0.05, ** = *p* < 0.01; *** = *p* < 0.001.

**Figure 3 healthcare-10-00776-f003:**
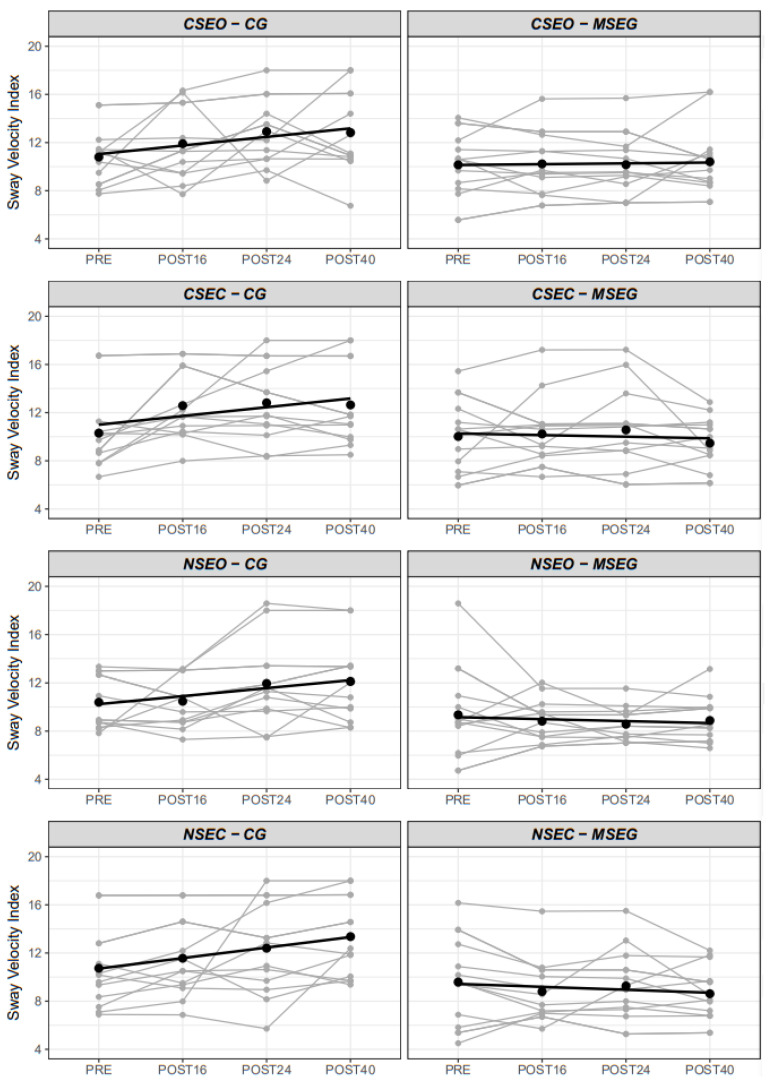
Individual and mean sway velocity index in the four conditions by the group. CSEO = Comfortable Stance Eyes Open; CSEC = Comfortable Stance Eyes Closed; NSEO = Narrow Stance Eyes Open; NSEC = Narrow Stance Eyes Closed.

**Table 1 healthcare-10-00776-t001:** Protocol for the muscular strength exercise program (MSE).

Warm-Up (Dynamic Flexibility and Walk around the Room): 5 min and RPE 3–5								
Exercises	Sets	Reps	Cadence	Resting Interval	RPE	Progression	Weeks	Intensity *
Front squat	2–3	10–20	2:3	30”	6 to 7	3 × 10^−15^	2	Yellow
Unilateral hip flexion (chair)	2–3	10–20	2:3	30”	6 to 7	3 × 15^−20^	2	Yellow
Row (with flexion) (chair)	2–3	10–20	2:3	30”	6 to 7	3 × 10^−15^	2	Red
Chest Press (stand/chair)	2–3	10–20	2:3	30”	6 to 7	3 × 15^−20^	2	Red
Reverse fly (stand/chair)	2–3	10–20	2:3	30”	6 to 7	3 × 10^−15^	2	Green
Shoulder Press/twist	2–3	10–20	2:3	30”	6 to 7	3 × 15^−20^	2	Green
Frontal raiser (stand/chair)	2–3	10–20	2:3	30”	6 to 7	3 × 15^−20^	2	Blue
Biceps curl (stand/chair)	2–3	10–20	2:3	30”	6 to 7	4 × 15^−20^	2	Blue
Overhead triceps extension	2–3	10–20	2:3	30”	6 to 7			
**Circuit format**								
Multidirectional walk around the room with an obstacle, cones, etc.				3–5 min	4 to 7			
Balance/ agility/motor coordinator exercises				3–5 min	4 to 7			
**Cooling down**								
Upper and Lower body’s static stretching (seated and standing)				5 min	2 to 3			

Reps = repetitions; RPE = Rating of Perceived Exertion of Borg Scale; min = minutes. * Based on Thera-band grade of elastic resistance.

**Table 2 healthcare-10-00776-t002:** Sample characteristics, postural control, and composite index outcomes at baseline (i.e., PRE).

Variables	MSEG ẋ ± SD	CG ẋ ± SD	*p*-Value
Total of participants Male Female	14 4 10	13 3 10	
Chronological age (years)	86 ± 3	87 ± 4	0.589
Height (cm)	155 ± 7.4	152 ± 10.2	0.389
Weight (kg)	70.4 ± 15.3	69.4 ± 11	0.845
Body mass index	29.1 ± 5.2	30 ± 4	0.616
Postural Control:	ẋ [95%CI]	ẋ [95%CI]	*p*-value
(i) CSEO	10.2 [8.6 to 11.7]	10.8 [9.2 to 12.4]	0.566
(ii) CSEC	10 [8.5 to 11.6]	10.3 [8.7 to 11.9]	0.801
(iii) NSEO	9.4 [7.8 to 10.9]	10.4 [8.8 to 12]	0.353
(iv) NSEC	9.6 [8 to 11.1]	10.7 [9.1 to 12.3]	0.308
Composite Index	10.1 [8.9 to 11.4]	10.5 [9.2 to 11.8]	0.665

Values are estimated marginal means with 95% confidence intervals (CI). Simple effect analysis is employed to obtain *p*-values. MSGE = Muscular Strength Exercise Group; CG = Control Group; ẋ = Mean; SD = Standard deviation; CSEO = Comfortable Stance Eyes Open; CSEC = Comfortable Stance Eyes Closed; NSEO = Narrow Stance Eyes Open; NSEC = Narrow Stance Eyes Closed.

**Table 3 healthcare-10-00776-t003:** Composite index (i.e., fall risk estimation) for both groups in each moment of measurement.

Condition and Moment	MSEG ẋ [95%CI]	MSEG ∆%	CG ẋ [95%CI]	CG ∆%
Composite Index—Baseline, PRE	10.1 [8.9 to 11.4]		10.5 [9.2 to 11.8]	
Composite Index—Training, POST16	9.3 [8.1 to 10.1]	−7.9%	12.3 [11 to 13.6]	17.2% *
Composite Index—Detraining, POST24	9.9 [8.7 to 11.1]	6.4%	12.6 [11.4 to 13.9]	2.4%
Composite Index—Retraining, POST40	9.6 [8.4 to 10.8]	−3.1%	12.9 [11.6 to 14.2]	2.3%

Values are estimated marginal means with 95% confidence intervals (CI). MSEG = Muscular Strength Exercise Group; CG = Control Group. * Significant differences in comparison to Baseline (*p* < 0.01).

## Data Availability

The data presented in this study are available on request from the corresponding author. The data are not publicly available due to [restrictions e.g. their containing information that could compromise the privacy of research participants].

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
