# Peer review of "The Effect of a Resistance Training, Detraining and Retraining Cycle on Postural Stability and Estimated Fall Risk in Institutionalized Older Persons: A 40-Week Intervention"

_healthcare, 2022, doi:10.3390/healthcare10050776_

Round 1
Reviewer 1 Report
The propensity toward instability and falling increases with age and is a widespread cause of morbidity and mortality amongst both the extremely aged- and younger individuals. This paper is an account of well-done and compelling research that, in a perfect world, should receive wide-spread attention in the media. It shows, beautifully, that even at advanced age, octogenarians may improve their balance through moderate resistance band-based exercise performed twice per week. It is striking that the amount of exercise required for improvement and protection can reasonably be incorporated into most person’s routines. As such, these findings may encourage individuals to continue being active into their later years, thereby increasing their quality of life-and decreasing healthcare burdens to themselves and society. What follows is purely in the spirit of improving the paper such that it be understandable to the general public.
The paper is beautifully written. A through edit for a minor number of small grammar mistakes is recommended. The Introduction and Methods are quite clear.
Line 60. 27,000?
Line 62. fall
Lines 69-70. Please supply reference
Line 89. This, taken together with the lack of work…
Lines 108 and 109. A general audience of readers may appreciate a bit more explanation of muscle power vs muscle strength, perhaps with examples of how to increase power development.
A question about Methods and Figure 1 caption: Did the CG group do any organized and repetitive activities (such as listening to preferred music, see 191) during the training periods for the MSE group. Or did they just go about their normal activities?
Lines 178-186. The description of the data collection procedures is quite good. Again, for a more general audience, a picture of the Physiosensing Fall Risk test, would be beneficial since most people have never seen one of these. In fact, if this journal permits video appendices, a video of this and the exercise protocols with the resistance bands would be a great addition.
Line 191. …choose their preferred music… This generated a pause and a smile. Is there perhaps more interesting data here? This reviewer of a certain age has a better work-out listening to ancient American/ British rock and roll. What do 85-yearold Portuguese persons prefer? Did it make a difference in their work-out and subsequent stability performance? Perhaps a subject for a future paper!
Table 1. MSE protocols. How many of these exercises targeted core muscle groups vs. extremities? One would think truncal musculature plays a more important role in maintaining balance and stability and may even have a larger proprioceptive representation in the dorsal column nervous system. True or not? Please add some commentary to Methods and/or Discussion explaining how these exercises were chosen and why.
Lines 285-297 seem to reflect numerous significant findings not reflected in the graphs by asterisks. An additional table of these findings might be effective.
Figure 3. Love this figure! It very effectively illustrates that, overall, the exercise group saw no further declines while the control group, overall, continued to decline. That there were protective effects during the detraining period is also very clear. Keep moving, people!
Lines 398-404. Rewrite for clarity.
Reviewer 2 Report
In the present review, Rodrigues et al. investigated the effects of resistance training, detraining, and retraining cycle on postural stability and estimated fall risk in older persons. Although this is quite an interesting and original paper manuscript needs revision.
I recommend that the text should be edited by a native English-speaking person. I suggest some changes below:
Introduction
The role of CNS in preventing/developing falls is missing.
Line 58
Authors reference to Behavioral Surveillance and Risk System from 2014. This data seems to be quite old.
Line 70
Authors wrote: “percentage of older people who fall increases with age…and it occurs mainly because of the degenerative process in muscle mass and function”. The role of the CNS seems to be as much important as the degenerative role of muscles.
Line 82
Authors should highlight the fall-preventing role of others forms of training- especially static forms and sensorimotor training.
Line 103
The benefits of exercise started to decline much faster than after 12 weeks of detraining.
Line 113
Please carefully check the definition of muscle power. The development of muscle power in older adults in the context of fall-preventing is not a good idea.
Material and methods
There is no effect size calculation. The authors should analyze the participant's history of falls as well. It is a crucial factor that can affect the outcomes.
Line 126
The study included only 27 participants. In this type of study, it is an insufficient number.
Line 136
Did the authors analyze the physical activity of participants? Study participants could not participate in organized physical activities but could have completely different individual physical activities.
Line 150
Two sessions a week? Why only 2?
226
The authors calculated only the body mass index. The body composition, especially muscle mass and fat mass, would be a much better parameter.
The authors should add a limitation paragraph.
Reviewer 3 Report
The authors are examining the effect of resistance training and detraining postural control among adults over the age of 80. Examining modifiable factors of falls among adults 80 is an important topic. There are a few critical concerns with the current analyses.
The authors are indicating that the resistance training attenuates age related declines in postural control. However if the decline noted among the control group from baseline to week 16 is age related then further declines would have been seen from week 16 to week 40. If this is the natural age related decline, can the authors explain why there was no detraining effect? Based on the current data in the context of the study design, it is not clear the investigators identified age related decline in postural control.
The within group analyses from table 3 did not show significant improvement in postural control. There appear to be large variations in data and no between group differences based on data from figure 2. The authors indicate significant between group differences, however examination of the figures indicate 3 individuals in the control group had poorer performance on the two of the postural control conditions. It is not clear that a majority of the control group had a reduction in scores. In fact, reviewing of the figure 3 it is difficult to determine any trend for either condition. There does not seem to be consistent evidence supporting the authors comments. The authors have a very small sample size which means each participant has a significant influence on group changes. A larger sample size is needed.
The WHO recommendations are for 150-300 minutes of moderate intensity physical activity, not light to moderate.
In Table 1, is there progression for triceps? The second half of the table is not clear. Description of the training program needs clarification. For example, what is meant by line 215 alternating muscle groups?
The paper needs extensive review for grammar and word choice. There are some examples where grammar needs to be corrected (line 44 should read maintaining health; line 63 highly costly price; line 148 if there is a first, there should be a second; line 210 colored soft-to-hard band) and other situation where the language needs to be aligned with scientific writing. For example, the literature uses the terms physical activity or exercise (please reference ACSM or WHO recommendations). Another example, on line 60 consider “are projected to increase 30% by 2030”. On line 354 what is meant by barely varied, was there a test to verify this?
Round 2
Reviewer 2 Report
Thank You for including all my suggestions.
Author Response
Thank the reviewer for the comments that lead to improving the quality of our work.
Reviewer 3 Report
Given the request for a 3-day review time, the paper was not read for grammar and syntax.
The authors reported a decline in function from the control group across the first 16 weeks, yet this trajectory was not sustained out to week 40 (see figure 2). As such, labeling it as age related does not appear appropriate. The paper hinges on the claim to reduce age related decline, when it is not clear they identified age related decline.
The authors provided power calculations. The authors did not address the likely influence of 2-3 individuals. As seen in figure 3, there appears to be 2-3 individuals in the intervention group improved, however there does not appear to be a group change. Based on the low sample size a few individuals have significant leverage. Providing a power calculation does not address the issue. Perhaps if sensitivity analyses were performed, removing these individuals and demonstrating the effect remains, a case could be made.
The claims of the paper appear hinged on 1) spurious change in the control group and 2) two to three intervention participants who improved. The original recommendation was to reject the article. There does not appear to be any additional information to change that perspective.
